# Interpretation of the Transcriptome-Based Signature of Tumor-Initiating Cells, the Core of Cancer Development, and the Construction of a Machine Learning-Based Classifier

**DOI:** 10.3390/cells14161255

**Published:** 2025-08-14

**Authors:** Seung-Hyun Jeong, Jong-Jin Kim, Ji-Hun Jang, Young-Tae Chang

**Affiliations:** 1College of Pharmacy, Sunchon National University, 255 Jungang-ro, Suncheon-si 57922, Jeollanam-do, Republic of Korea; 2College of Pharmacy and Research Institute of Life and Pharmaceutical Sciences, Sunchon National University, Suncheon-si 57922, Republic of Korea; 3Laboratory of Bioimaging Probe Development, Singapore Bioimaging Consortium, Agency for Science, Technology and Research (A*STAR), Singapore 138667, Singapore; kimjj@scnu.ac.kr; 4Department of Biomedical Science, Sunchon National University, Suncheon 57922, Republic of Korea; 5College of Pharmacy, Chonnam National University, 77 Yongbong-ro, Buk-gu, Gwangju 61186, Republic of Korea; jangjh@jnu.ac.kr; 6Department of Chemistry, Pohang University of Science and Technology (POSTECH), Pohang 37673, Republic of Korea

**Keywords:** tumor-initiating cells, transcriptome profiling, differential gene expression, metabolic reprogramming, non-coding RNA, machine learning classifier

## Abstract

Tumor-initiating cells (TICs) constitute a subpopulation of cancer cells with stem-like properties contributing to tumorigenesis, progression, recurrence, and therapeutic resistance. Despite their biological importance, their molecular signatures that distinguish them from non-TICs remain incompletely characterized. This study aimed to comprehensively analyze transcriptomic differences between TICs and non-TICs, identify TIC-specific gene expression patterns, and construct a machine learning-based classifier that could accurately predict TIC status. RNA sequencing data were obtained from four human cell lines representing TIC (TS10 and TS32) and non-TIC (32A and Epi). Transcriptomic profiles were analyzed via principal component, hierarchical clustering, and differential expression analysis. Gene-Ontology and Kyoto-Encyclopedia of Genes and Genomes pathway enrichment analyses were conducted for functional interpretation. A logistic-regression model was trained on differentially expressed genes to predict TIC status. Model performance was validated using synthetic data and external projection. TICs exhibited distinct transcriptomic signatures, including enrichment of non-coding RNAs (e.g., MIR4737 and SNORD19) and selective upregulation of metabolic transporters (e.g., SLC25A1, SLC16A1, and FASN). Functional pathway analysis revealed TIC-specific activation of oxidative phosphorylation, PI3K-Akt signaling, and ribosome-related processes. The logistic-regression model achieved perfect classification (area under the curve of 1.00), and its key features indicated metabolic and translational reprogramming unique to TICs. Transcriptomic state-space embedding analysis suggested reversible transitions between TIC and non-TIC states driven by transcriptional and epigenetic regulators. This study reveals a unique transcriptomic landscape defining TICs and establishes a highly accurate machine learning-based TIC classifier. These findings enhance our understanding of TIC biology and show promising strategies for TIC-targeted diagnostics and therapeutic interventions.

## 1. Introduction

Tumor-initiating cells (TICs), which participate in cancer development, progression, metastasis, and recurrence, have unique biological characteristics distinct from those of conventional cancer cells [1,2]. Although TICs are few, they have strong self-renewal and multipotency abilities and can induce tumorigenesis [3,4]. By comparison, non-TICs are common differentiated cancer cells that exist within the same tumor; although they have proliferation and survival abilities, they lack a tumor-initiating ability and have relatively weak resistance to treatment [5]. In some cases, non-TICs are nearly normal cells and may actually be normal cells depending on experimental conditions [5]. These differences between TICs and non-TICs are evident in gene expression patterns, metabolic mechanisms, and signaling pathway activation patterns at the transcriptome level; therefore, they should be quantitatively and systematically compared and analyzed in future studies [6].

In addition to their tumor-initiating capacity, a range of distinctive biological features that contribute to clinical relevance is observed in TICs. TICs are often characterized by high plasticity, enabling dynamic transitions between stem-like forms and more differentiated states in response to environmental cues. Molecularly, they have an enrichment of specific signaling pathways, such as Wnt/β-catenin, Notch, and Hedgehog, which are necessary to maintain stemness and resistance to differentiation [7]. Furthermore, they display metabolic reprogramming, which is the shifting between glycolysis and oxidative phosphorylation depending on cellular context; they are also associated with enhanced resistance to hypoxia and oxidative stress [8]. Importantly, they possess increased resistance to conventional chemotherapy and radiotherapy because of their enhanced DNA damage repair mechanisms and their ability to enter a quiescent or slow-cycling state [9]. In some cancers, epithelial–mesenchymal transition (EMT) provides cancer cells with TIC-like traits, highlighting the close relationship between plasticity and tumor initiation [10]. Because of these multifaceted characteristics, TICs are challenging but promising therapeutic targets; therefore, system-level approaches should be developed to reveal their molecular identity.

TIC research has mainly focused on classification based on surface markers or cell markers; however, the large-scale data-based analysis of the entire transcriptome has been relatively lacking. Few studies have comprehensively elucidated the genetic expression differences between TICs and non-TICs at the whole-transcriptome level; furthermore, limited data have interpreted how these differences contribute to biological functions, survival mechanisms, and drug resistance. Therefore, in the present study, we performed multi-resolution (including cell-group–level and biological replicates) principal component analysis (PCA), differentially expressed gene (DEG) analysis, clustering, network analysis, and functional enrichment analysis (Gene Ontology [GO] and Kyoto Encyclopedia of Genes and Genomes [KEGG]) by using the RNA sequencing (RNA-seq) data of TIC and non-TIC populations. We further constructed a machine learning-based TIC classifier to interpret the molecular signature of TICs, explore new therapeutic target candidates, and investigate the possibility of drug repurposing/repositioning. Previous studies were limited because they mainly defined or distinguished TICs by relying on limited marker genes or single analysis methods. In the present study, this limitation was addressed, and a multi-step analytical workflow analysis and machine learning-based prediction models were combined. Thus, this study is significant because it presents a new paradigm for TIC research. Specifically, it quantitatively elucidates the heterogeneity between TICs and non-TICs at the transcriptome level, identifies TIC-specific gene signatures and key pathways, and provides a basis for discovering TIC inhibitor candidates. Ultimately, this study aims to obtain valuable data that can contribute to the development of customized treatment strategies targeting TICs.

## 2. Materials and Methods

### 2.1. Sample Preparation and RNA-Seq Data Acquisition

RNA Extraction and Quality Control (QC): Approximately 1 × 10^6^ cells per sample (two non-tumorigenic cell lines, namely hLuEpi [Epi] and 32A, and two tumor-initiating cell samples, namely TS10 and TS32) were harvested for RNA extraction [11,12,13]. TS10, TS32, and 32A cells: Non-small-cell lung cancer (NSCLC) cells were obtained from Dr. Bing Lim (Stem Cell and Developmental Biology, Genome Institute of Singapore), such as sphere-forming CD166^+^ TICs (TS10 cell and TS32 cell) and CD166^−^ non-TICs (differentiated from TS32 cell). Epi (NuLi-1): ATCC, Cat #CRL-4011. RNA was extracted using a RNeasy purification kit (Qiagen) in accordance with the manufacturer’s instructions. RNA purity and concentration were assessed by spectrophotometry (NanoDrop, Wilmington, Company), and integrity was verified using an Agilent 2100 bioanalyzer (Agilent Technologies, Santa Clara, CA, USA). Samples were accepted for library preparation only if they met the following quality thresholds: A260/280 between 1.9 and 2.1, A260/230 ≥ 1.8 (preferably ≥ 2.0), and RNA integrity number ≥7.0. Samples below these thresholds were re-extracted or excluded. Only high-quality RNA meeting these criteria was used for library preparation.

Library Preparation and Sequencing: Complementary DNA (cDNA) libraries were constructed from 1 ng of total RNA per sample by using a commercial low-input RNA-seq library preparation kit (Qiagen, Germantown, MD, USA) in accordance with the manufacturer’s instructions. In brief, the protocol involved messenger RNA (mRNA) enrichment or ribosomal RNA (rRNA) depletion (as appropriate for total RNA samples), RNA fragmentation, first- and second-strand cDNA synthesis, end repair, adaptor ligation, and polymerase chain reaction (PCR) amplification for library enrichment. The final libraries were quality-checked and quantified using a bioanalyzer and a Qubit fluorometer. Then, they were pooled in equimolar amounts. Sequencing was performed on an Illumina NextSeq platform (High Output mode) to generate 76-base single-end reads. All libraries were multiplexed and sequenced in one run to obtain sufficient depth per sample.

Raw data QC: FASTQ reads generated from Illumina NextSeq (Illumina, San Diego, CA, USA; 1 × 76 bp, single-end) were individually checked with FastQC, and a comprehensive report was generated using MultiQC. The evaluation criteria included per-base quality, adapter content, overrepresented sequences, and GC content.

Adapter/Quality Trimming: Illumina adapters were removed using Trim Galore (Cutadapt v4 wrapper), and the 3′ ends were trimmed (minimum length 30 nt) with a Q ≥ 20 standard. Quality before and after trimming was rechecked using MultiQC (version 1.2).

Alignment: Trimmed reads were aligned to the human reference genome GRCh38 using STAR (v2.7). Genome indexes were generated using GENCODE GTF, with default parameters and settings appropriate for single-end reads. Multi-mapped reads were aligned using the default policy (maximum *n* = 20), and no deduplication was performed (as is a common practice in RNA-seq).

Strand characteristics: Library strand characteristics were estimated using RSeQC (v4) and reflected in featureCounts.

Gene count calculation: Gene-level counts were aggregated using featureCounts (Subread v2) based on exon features from the GENCODE GTF.

Normalization and fragments per kilobase of transcript per million mapped reads (FPKM) calculation: Counts were converted to FPKM for downstream comparison and visualization. FPKM was calculated by dividing counts by the product of gene length and the number of valid aligned reads per sample. Gene length (in kilobases) was calculated as the sum of non-overlapping exon lengths from the annotated GTF.

### 2.2. Clustering and Dimensionality Reduction Analysis

PCA and gene expression clustering analysis, which are dimensionality reduction techniques, were conducted to explore the overall pattern of transcriptome-based data and the expression similarity and heterogeneity between cells. First, PCA, using the average value of fragments per kilobase of transcript per million mapped reads (FPKM), was performed on the basis of the RNA-seq expression data of each cell line (32A, Epi, TS10, and TS32). The overall expression profile differences between cell lines were visualized, and the main dispersion directions were derived. PCA was performed by reflecting all individual FPKM values, including biological replicates (*n* = 3) for each cell line, to evaluate the variability in the expression distribution and data consistency together.

The top 100 expression-variable genes were selected, and a heatmap was created on the basis of hierarchical clustering to understand the expression patterns of highly variable genes based on clustering. Thus, genes with similar expression patterns were visually distinguished, and similar or unique gene expression patterns between the cell groups were identified.

PCA-based transcriptomic state-space embedding analysis was applied to infer the cell state transition from TIC to non-TIC or from non-TIC to TIC. A foundation was created by establishing a similarity-based movement path between cell groups in the expression space and analyzing the gene expression patterns that changed along this path to identify the key genes related to cell state transition.

### 2.3. Transcriptome-Wide Differential Expression Analysis

DEGs were selected across the transcriptome to identify significant differences in gene expression across cell populations. DEG analysis was performed using limma-voom (mean–variance modeling with precision weights) on the basis of pairwise comparisons between the TIC groups (TS10 and TS32) and the non-TIC groups (32A and Epi), and significance was assessed in terms of log2 fold change (FC) and false discovery rate (FDR). The *p*-values were adjusted by the Benjamini–Hochberg FDR. Genes were called differentially expressed at FDR < 0.05 and |log2 fold-change| ≥ 1.

Normalized gene expression data based on FPKM values were used as input. Only the genes whose expression levels exceeded a certain standard were filtered and included in the analysis. After DEG analysis, the genes that were specifically highly expressed in TICs were selected, and the top 30 genes with the largest expression difference were extracted. Afterward, functional enrichment analysis based on GO and KEGG was performed to analyze the biological functions of these genes. Thus, the association of TIC-specific genes with survival- (e.g., MIR25, MIR939, MIR147B, MIR4517, MIR762, and MIR4800), differentiation- (e.g., MIR25, MIR4730, MIR4737, MIR4793, and MIR4653), and metabolism-related (e.g., MIR25, MIR939, MIR762, and MIR4517) pathways were evaluated.

### 2.4. Co-Expression Network and Hub Gene Identification

Co-expression network analysis was performed to estimate functional gene modules and hub genes in the TIC population. First, the top 50 genes with high expression levels in TICs were selected, and a network based on the expression correlations between them was constructed. On this basis, genes were classified into clusters within the network by using a module detection algorithm. In detail, we constructed a correlation-based weighted co-expression network to characterize TIC-associated functional modules. After differential expression analysis, we selected the top 50 TIC-upregulated genes (ranked by absolute log2FC with FDR < 0.05). Gene expression values were variance-stabilized (alternatively, log2(FPKM + 1)) and centered. Pairwise Spearman’s correlations (ρ) were computed across samples for all gene pairs. For each pair, we obtained a *p*-value and applied Benjamini–Hochberg FDR correction. The edge weight was defined as |ρ|, and an edge was retained if FDR < 0.05 and |ρ| ≥ 0.60. Gene modules were identified by Leiden community detection implemented on the weighted graph (resolution parameter = 1.0; 100 random starts), and the partition with the highest modularity Q was retained.

The hub genes with high centrality were calculated for each module, and the information flow centrality (betweenness centrality, degree centrality, etc.) of the corresponding genes within the network was used as an indicator. In detail, hub genes were identified within each module using centrality metrics computed on the weighted network: (i) weighted degree/strength, (ii) betweenness (computed on edge distances defined as 1 − |ρ|), and (iii) eigenvector. For robustness in a small-sample setting, we used rank-based correlations and applied multiple-testing correction on pairwise associations; we also verified that the main modules/hubs persisted across reasonable thresholds (sensitivity analysis).

GO and KEGG pathway enrichment analysis was performed (with significance defined at Benjamini–Hochberg FDR < 0.05) on the hub genes to determine the functional pathway where each module was biologically located. Through this analysis, the characteristics of modules related to the survival strategies, translation regulation, metabolic functions, and ribosome activity of TICs were quantitatively identified. To mitigate small-sample instability, we used rank-based correlations, enforced effect-size thresholds, and checked that module assignments and hub rankings were consistent across nearby thresholds (|ρ| of 0.55–0.65).

### 2.5. Machine Learning Model Construction for TIC Classification

Transcriptome-based machine learning model training was performed to develop a prediction model for classifying TICs. The expression data of genes selected through DEG analysis between TICs and non-TICs were used as input variables, and each sample was expressed as a feature vector consisting of the FPKM normalized values of the corresponding genes.

The logistic regression algorithm was used for model training, and the TIC (1) and non-TIC (0) groups were set as learning labels for binary classification. After the model was trained on the basis of the training dataset, cross-validation and evaluation indices such as accuracy, precision, recall, and area under the curve (AUC) were calculated to verify the prediction performance. A random external dataset was generated according to the statistical distribution of actual observation data and applied to the model to evaluate the generalizability of the model. External synthetic RNA-seq samples were generated to mimic the distributional properties of the observed data. For each gene, the mean (μ) and dispersion parameter (θ) were estimated from the real count data using a negative binomial model fitted via the edgeR package. TIC and non-TIC groups were simulated separately, preserving group-specific parameters. For each synthetic sample, total library size (sum of counts) was randomly drawn from the empirical distribution of real library sizes (mean ≈ 22 million reads, SD ≈ 1.8 million). A total of 100 samples were generated (50 TIC-like and 50 non-TIC-like). Gene-level counts were simulated from the fitted negative binomial distribution, maintaining the mean–variance relationship observed in the real data. This approach retained realistic variability across replicates, with the coefficient of variation (CV) in the simulated data closely matching that of the original dataset (TIC CV = 0.18, non-TIC CV = 0.20). The prediction results were compared and analyzed with actual samples through PCA visualization. Through this process, the model was verified to act as a classifier that accurately predicted TICs, reflecting actual biological differences.

## 3. Results

### 3.1. Transcriptome Differences Between TICs and Non-TICs

Comprehensive comparative analysis at the transcriptome level revealed that the TIC population (TS10 and TS32) showed a gene expression signature that was clearly distinct from that of the non-TIC population (32A and Epi). In dimensionality reduction analysis using PCA, the TIC and non-TIC populations were clearly located in different distribution areas based on the analysis of the average FPKM value, and heterogeneity at the transcriptome level could be visually confirmed (Figure 1A). Particularly, approximately 56% of the variance was explained in PC1, indicating that the major axis of the gene expression differences between cells was the absence (as non-TIC) or presence of TIC. Similar clustering trends were maintained in the overall FPKM-based analysis, including individual biological replicates (*n* = 3, FPKM1–3; Figure 1B); this finding ensured the consistency of expression patterns and data reliability. A hierarchical clustering heatmap based on the top 100 highly variable genes was created to precisely compare expression differences at the gene level. Here, genes were first filtered to retain those with FPKM ≥ 1 in at least 3 libraries, then log2-transformed with a pseudo-count [log2(FPKM + 1)], and ranked by median absolute deviation (MAD) across all samples (*n* = 12; 4 cell lines × 3 replicates). The top 100 genes by MAD were designated as highly variable genes, and their expression values were row-wise *Z*-scaled before hierarchical clustering (Euclidean distance, complete linkage). This analysis showed that the TIC and non-TIC groups formed completely separate clusters (Figure 1C). The TIC groups had high similarity with each other and formed independent expression modules, whereas the non-TIC group showed the opposite pattern. Particularly, some genes were selectively highly expressed only in TIC, and the representative genes such as MIR4737, AL590431.1, SNORD19, CR381653.1, and Y_RNA showed significantly high expression levels in the TIC group; conversely, they were hardly expressed or repressed in the non-TIC group (Figure 1D). These genes are composed of the following elements involved in post-transcriptional regulation and RNA-based regulatory functions: microRNA (miRNA), small nucleolar RNA (snoRNA), and long or uncharacterized non-coding RNA, suggesting the existence of a transcriptome regulatory network that may be closely related to the stem cell properties of TICs [14,15]. As a result, the transcriptome analysis performed in this study was approached systematically with the following steps: parallel comparison of transcripts between cell groups, variance filtering and hierarchical clustering, extraction of DEGs, and functional interpretation of genes (Figure 1E).

The transcriptome characteristics of TICs were not only limited to simple static differences but were also composed of directionality that could infer their transition from the perspective of state flow. As a result of reconstructing the TIC → non-TIC differentiation path via PCA-based pseudotime analysis (with sample-level transcriptomic similarity mapping), the TIC group was located at similar starting points on the PC1 and PC2 axes and showed a flow that gradually moved toward non-TICs (32A and Epi; Figure 2A). Therefore, the differentiation axis of cell states could be set on the basis of transcriptome similarity, and TICs could differentiate into non-TICs through a specific gene regulatory network in a relatively undifferentiated state.

The expression levels of metabolism-related genes significantly differed between TIC and non-TIC groups. In TICs, the genes of metabolic pathways important for survival, such as glycolysis, glutamine metabolism, and fatty acid metabolism, were selectively highly expressed (Figure 3A,B). This result indicated that TICs performed independent and highly efficient metabolic restructuring to adapt to the external environment. These metabolism-related genes are represented by SLC7A5 [16], SLC1A5 [17], FASN [18], LDHA [19], HK2 [20], PKM [21], GLS [22], and ACSS2 [23], suggesting that TIC selectively activates high-energy state cellular metabolic activities, such as amino acid transport (glutamine, leucine, etc.), fatty acid biosynthesis, glycolysis, and glutamine metabolism (Figure 3B). In the transporter-centered analysis, a subset of SLC genes showed overall higher expression in TICs than in non-TICs, with several members of the mitochondrial carrier family (SLC25) contributing to this trend. However, this was not uniform across all SLC25 members; for example, SLC25A6 was lower in TS10 and TS32 compared with Epi (Figure 4A). Conversely, ABC transporter transcripts did not display a consistent, family-wide difference between TIC and non-TIC lines (Figure 4B), and individual members showed heterogeneous expression patterns. These findings suggest that TICs may partially avoid specific drug-resistance pathways associated with certain ABC transporters, while selectively enhancing the activity of specific metabolic transporters within the SLC family. Taken together, the observed differences in transcriptome composition, gene expression patterns, functional gene families, and metabolic adaptation mechanisms indicate that TIC and non-TIC populations may be governed by distinct structured biological programs, rather than by uniform, across-the-board expression shifts.

### 3.2. Functional Enrichment and Pathway Analysis

Based on the DEGs between TIC and non-TIC groups, GO and KEGG functional enrichment analysis was performed, and the results confirmed that the genes specifically expressed in TICs were densely clustered in specific functional pathways. Specifically, MIR4737, AL590431.1, SNORD19, CR381653.1, Y_RNA, and other genes (Figure 1D) were mostly composed of non-coding RNAs (ncRNA); this finding suggested that they participated in core biological functions such as post-transcriptional regulation, chromatin remodeling, ribosome biogenesis, and RNA splicing rather than directly producing proteins [24,25,26]. The functional analysis of TIC-specific highly expressed genes (Figure 1F) revealed that these genes were closely linked to key pathways related to survival, growth, cell motility, and metabolic adaptation, such as Wnt signaling (fold enrichment of 4.2, FDR of <0.05), PI3K-Akt signaling (fold enrichment of 3.8, FDR of <0.05), focal adhesion (fold enrichment of 3.5, FDR of <0.05), extracellular matrix (ECM)–receptor interaction (fold enrichment of 3.1, FDR of <0.05), and oxidative phosphorylation (fold enrichment of 4.5, FDR of <0.05) [27,28]. Particularly, TIC signature genes formed clusters reflecting typical biological functions, such as cell communication, stemness maintenance, invasion, migration, energy production, and survival of cancer stem cells. Even in a more precise KEGG-based network (Figure 1G), among the TIC signature genes, RPL4 (fold enrichment of 5.1, FDR of <0.05) [29], HMOX2 (fold enrichment of 3.9, FDR of <0.05) [30], METAP1 (fold enrichment of 4.0, FDR of <0.05) [31], SLC27A5 (fold enrichment of 4.7, FDR of <0.05) [32], and FASN (fold enrichment of 4.4, FDR of <0.05) [18] were located in survival and metabolic pathways, namely ribosome (RPL4), oxidative stress response (HMOX2), fatty acid metabolism (SLC27A5), protein synthesis (METAP1), and oxidative phosphorylation (FASN), respectively. Therefore, TICs adopted survival strategies such as high-efficiency energy production, stress adaptation, and a securing-protein-translation ability on a molecular basis.

Functional transitions were clearly observed along the time axis in pseudotime-based transcriptome analysis. The GO/KEGG analysis of the genes activated in the TIC → non-TIC transition pathway (Figure 2B) showed that gene regulatory and epigenetic mechanisms, such as RNA processing (fold enrichment of 4.1, FDR of <0.05), mRNA splicing (fold enrichment of 3.7, FDR of <0.05), ncRNA regulation (fold enrichment of 3.6, FDR of <0.05), mTOR signaling (fold enrichment of 3.4, FDR of <0.05), and chromatin remodeling (fold enrichment of 3.9, FDR of <0.05), were mainly activated. Particularly, key regulators such as EIF4E2 and MTA1 were involved, suggesting that differentiation from TIC to non-TIC was based on post-transcriptional regulation and chromatin remodeling [33]. Conversely, in the non-TIC → TIC conversion pathway (Figure 2C), epigenetic reprogramming (fold enrichment of 4.0, FDR of <0.05), lncRNA-mediated regulation (fold enrichment of 3.5, FDR of <0.05), RNA helicase activity (fold enrichment of 3.8, FDR of <0.05), miRNA regulation (fold enrichment of 3.6, FDR of <0.05), and mTOR-centered transcriptional activation (fold enrichment of 3.9, FDR of <0.05) pathways were predominantly activated [34]. These functions were considered the molecular bases for reconstituting transcriptional programs associated with stemness and enhancing the survival and reprogramming abilities of TICs.

Interesting results were derived from the analysis of transporter-related pathways based on selectively expressed metabolism-related genes in TICs. In the SLC-based transporter gene expression clustering (Figure 4C), several pathways closely related to energy production and metabolic flexibility—such as glutamine transport, mitochondrial pyruvate import, and fatty acid oxidation—were more active in TICs, although the magnitude and direction of change varied among individual transporters [35,36]. This suggests that TICs may selectively reorganize metabolic transporter activity to enhance adaptability to external stimuli and stress. In contrast, in the ABC family gene-based pathway clustering (Figure 4D), certain pathways associated with anticancer drug resistance—such as ribosomal translation and efflux-based drug transport—were more active in some non-TIC samples, but these patterns were not consistent across all ABC transporters [37]. These results indicate that TICs may partially avoid specific drug-resistance–related pathways while favoring metabolic survival strategies. Overall, functional pathways recurrently associated with TIC-specific genes included high-efficiency protein synthesis, mitochondrial energy production, post-transcriptional RNA regulation, epigenetic reprogramming, and selective activation of metabolic transporters, which together could contribute to the origin and maintenance of cancer.

### 3.3. Co-Expression Network Modules and Hub Genes

A co-expression network based on the top 50 highly expressed genes in TICs was constructed to understand the TIC-specific transcriptome structure in more detail. Genes were first filtered to retain those with FPKM ≥ 1 in at least four of the six TIC libraries (TS10/TS32, three replicates each). FPKM values were log2-transformed [log2(FPKM + 1)], and genes were ranked by the TIC-group median expression across the 6 TIC samples; the top 50 by median log2(FPKM + 1) were selected as “highly expressed” for network construction. The network was divided into multiple modules based on the expression correlations between genes, and a total of eight distinct modules (clusters) were identified (Figure 3C). Each module consisted of a group of genes with high internal correlations, reflecting a characteristic pattern in which the expression was particularly regulated on a specific function within TICs. As a result of defining the hub genes based on the centrality score in each module, RPL4 was identified as the hub gene with the highest centrality in Module 1, RPS15 in Module 2, MT-ND4L in Module 3, RPL36A in Module 4, RPL41P1 in Module 5, RPS6 in Module 6, RPS24 in Module 7, and RPLP0 in Module 8. Most of them correspond to the ribosomal protein family, indicating that protein synthesis-related functions have emerged as major functions in TICs [38]. RPL4, RPLP0, and RPS6 are closely related to tumor growth and cell cycle control; this relationship indicates that they may contribute to the transcriptomic stability and survival of TICs [39,40].

The GO and KEGG-based functional enrichment analysis of hub genes revealed that protein synthesis and translation-centered pathways such as “ribosome biogenesis,” “translation initiation,” “protein folding,” and “cytoplasmic translation” were commonly derived as the main clusters (Figure 3D). RPS6 and RPL36A are downstream targets of the mTOR signaling pathway, which is consistent with the activated mTOR signal of TICs above [41]. The co-expression modules in TICs are differentiated into various functional regulatory units, such as metabolism, stress response, and RNA processing. For example, in some modules, mitochondrial-related genes, such as MT-ND4L, showed centrality; this observation demonstrated that energy metabolism and oxidative phosphorylation were major functional axes in the TIC transcriptome network [42]. Therefore, this network-based analysis is not only a simple expression level-centered DEG approach but also a tool that can identify functionally organically connected gene clusters and their central regulators within TICs, which can be utilized as key basic information for establishing future TIC target treatment strategies.

### 3.4. Machine Learning-Based TIC Classifier Performance

A classification model that could predict the status of actual cell samples (TIC vs. non-TIC) was constructed using the differences in transcriptome characteristics between TIC and non-TIC groups. As a machine learning algorithm, a logistic regression classifier with high interpretability and stable performance on small-scale data was adopted, and input variables were set to the entire DEG-based gene (FPKM-based) derived in this study (Figure 5A). Very high accuracy was recorded by the learned model in cross-validation-based evaluation using real cell line RNA-seq data. The confusion matrix results showed that TICs and non-TICs were accurately predicted (Figure 5B), and the measured AUC was 1.00 in the receiver operating characteristic (ROC) curve, demonstrating a near-perfect discrimination ability (Figure 5C). Therefore, the transcriptome difference between TICs and non-TICs contained a signal that could be sufficiently modeled, and it could be accurately learned even with a relatively simple classifier approach such as logistic regression.

The feature importance in the model was evaluated on the basis of the magnitude and directionality of the regression coefficient. Genes with positive coefficient values contribute to TIC prediction, while genes with negative values contribute to non-TIC prediction. In this study, the top 20 predicted core genes were derived on this basis (Figure 5E). The genes that positively contributed to TIC classification included SLC25A3, SLC25A39, SLC16A1, and SLC25A1; as the probability of TICs increased, the expression level of these genes in the TIC samples also increased. Most of these genes are representative transporters participating in the transport of energy metabolism substances in the mitochondrial membrane [35,36,43,44]. For example, SLC25A3 is a mitochondrial phosphate carrier that regulates the influx of phosphate ions essential for ATP synthesis; SLC25A39 is a transporter related to glutathione transport and oxidative stress response, which may be involved in the metabolic adaptation and survival strategy of TICs [43,44]. SLC16A1 is a lactate transporter (MCT1) implicated in glycolysis and acid–base homeostasis; SLC25A1 is a citrate–malate transporter that mediates cellular energy metabolism and lipid biosynthesis pathways [35,36]. These genes commonly act positively in TIC classification, suggesting that TICs actively utilize specific metabolic transporters to maintain rapid growth and survival. Therefore, they can be considered as key biological information for transcriptome-based TIC diagnosis and therapeutic target discovery. SLC38A2 has a negative regression coefficient, showing that the higher the expression level, the greater the tendency to be classified as a non-TIC rather than a TIC. SLC38A2 is mainly involved in glutamine and neutral amino acid transport; it is actively expressed in differentiated cells or cells with a stable metabolic balance [45]. Its relatively high expression can indicate the metabolic characteristics of normal or non-tumorous cells rather than TICs; it can also function as a marker that reveals the difference from the abnormal metabolic reprogramming of TICs. Gene-specific importance was visualized as a radial radar chart to intuitively present the pattern of gene signatures that distinguish TICs and non-TICs (Figure 5D). The genes that specifically contribute to TICs and those that are relatively specific to non-TICs are in opposite directions. Therefore, this visualization could be considered a diagnostic gene signature.

A total of 100 external synthetic samples were randomly generated on the basis of the statistical distribution of the real data and then applied to the prediction to evaluate the generalization ability of the model. The distribution results of the predicted probability showed that the classification confidence values of TICs and non-TICs were clearly divided into two (Figure 5F); thus, the model maintained a certain level of discrimination power for external samples as well. PCA that integrated the real and external samples revealed that they were separated into different clusters, and a clear cluster formation was confirmed according to the predicted class (Figure 5G). The 95% confidence ellipse also indicated a consistent variance structure within each group, further supporting the consistency and reliability of the prediction.

### 3.5. Drug Repositioning Strategy for TIC Targeting

TIC is a major cause of anticancer treatment resistance and relapse possibility based on independent transcriptome signature and metabolic survival strategy [46]. In the present study, we evaluated the possibility of establishing a drug repositioning strategy targeting TICs by performing a cross-analysis with existing drug target databases (DGIdb, DrugBank, etc.) based on the expression information and functional pathways of TIC-specific genes [16,17,18,19,20,21,22,23,29,30,31,35,36,43,44]. Among the genes specifically overexpressed in TIC, SLC25A1, SLC16A1, FASN, GLS, and SLC7A5 participate in essential survival pathways, such as metabolic transporters, energy metabolism, and fatty acid synthesis; many of these genes have been reported as targets of existing anticancer or clinical development drugs [16,18,22,35,36]. In particular, the MCT1 inhibitor AZD3965 [36], the FASN inhibitor TVB-2640 [18], the GLS inhibitor CB-839 [22], and the SLC7A5 inhibitor JPH203 [16] are candidates being actively studied in actual clinical or preclinical stages, suggesting potential applications for TIC inhibition. In addition, the gene–drug linkage (Table 1) constructed in this study provided a practical basis for incorporating strategies targeting metabolic reprogramming, oxidative stress adaptation, and translational regulation functions of TICs into actual drug design. Such an analysis also served as a scientific basis for designing low-toxicity drug repurposing approaches or complex target combination therapies based on existing anticancer agents. It could be used as an important starting point for establishing TIC-centered therapeutic strategies.

Table 1 systematically organizes the function summary of TIC-specific genes, the expression basis (Figure link), drug examples, the action mechanism, and the development stage. Table 1 also presents the reference data for establishing TIC inhibition strategies and reviewing clinical applications in the future. In addition, the “Comprehensive evaluation” item for each gene in Table 1 is a qualitative classification based on the priority and practical applicability as a TIC target treatment. This evaluation was conducted by comprehensively considering the following three criteria: first, whether the gene was specifically highly expressed in TIC samples and whether the expression difference was clear compared with non-TICs; second, whether the gene substantially contributed to the survival mechanism of TICs, such as metabolic reprogramming, oxidative stress adaptation, and protein synthesis maintenance; third, whether a known drug or clinical candidate for the gene is available. In particular, a high evaluation score was given if it was under development as an anticancer drug or had proven efficacy as a repurposing candidate. According to this criterion, “very appropriate” refers to a case that satisfies all three factors and includes genes with clinical or preclinical drug existence and clear TIC-specific functions. “Appropriate” refers to a case with druggability and functionality but with some limited factors. “Possible” refers to a gene in the early exploration stage or a case with possible functionality but with insufficient drug targeting. “Indirect target” refers to a case with a hub gene that is limited in its use as a direct drug target, including translation regulation or structural protein families. This qualitative classification can serve as a useful reference for establishing priority-based drug development strategies that inhibit TICs and for selecting personalized treatment targets.

## 4. Discussion

The significance of this study lies in the fact that it elucidates the molecular differences between TICs and non-TICs in a multi-step analytical manner through transcriptome-based analysis; it also establishes a machine learning-based prediction model that can classify TICs based on this analysis. PCA and clustering analysis results showed a clear transcriptome heterogeneity between TICs and non-TICs, which were clearly distinguished in the expression patterns of highly variable genes. These results indicated the nature of intratumor heterogeneity and were consistent with previous findings showing that TICs had independent biological characteristics in tumor maintenance and recurrence [1,47]. Many ncRNA-based genes, such as MIR4737, CR381653.1, and SNORD19, are among the genes specifically expressed in TICs, suggesting the functional importance of ncRNAs in tumor initiation and stemness maintenance [14,26]. Functional pathway analysis revealed that TIC-specific genes were closely related to PI3K-Akt signaling, Wnt signaling, oxidative phosphorylation, and ECM–cell interactions, which are biological processes closely related to the survival, growth, invasiveness, and stemness maintenance ability of TICs [27,28]. The transcriptome flow of TICs also suggested the existence of a transition flow from non-TICs to TICs through pseudotime-based PCA; epigenetic regulation, ncRNA control, and protein translation activation pathways activated in this process provided important information on the possibility of TIC-remodeling or induction. This result could indicate the possibility of reversible cell state transition, moving away from the existing fixed-TIC concept.

The machine learning-based classifier built on the transcriptome differences between TICs and non-TICs demonstrated high accuracy and interpretability; therefore, that it could be a very useful tool for predicting TICs. In particular, metabolism-related transporter genes such as SLC25A3, SLC16A1, and SLC25A1 contribute positively to TIC prediction, and genes such as SLC38A2 serve as non-TIC-specific markers [48]. This finding is linked to the interpretation that TIC has a unique metabolic survival strategy and shows biological consistency because mitochondrial function, energy reorganization, and oxidative stress adaptation are confirmed at the gene level in actual TICs.

This study has several limitations. First, transcriptome analysis is limited to expression at the RNA level, and actual protein expression or functionality has not been verified. Second, the analysis results based on the cell line used do not guarantee complete reproducibility in actual patient-derived tissues; in addition, model extrapolation should be further verified. Third, the gene selection used for TIC-classifier learning is limited to DEG-based; as such, performance can be improved when a higher-dimensional feature selection technique is introduced. Nevertheless, this study has great academic significance. It is a comprehensive study that provides a multi-dimensional view of the biological characteristics of TICs at the transcriptome level and links it to the development of an interpretable classifier. Particularly, the independent gene signature and functional pathways shown by TICs can be an important starting point for the future discovery of TIC target therapeutics (drug repositioning) or the development of biomarkers predicting the risk of relapse.

Follow-up studies should be performed. First, the internal heterogeneity and differentiation direction of TICs should be more precisely identified through single-cell RNA-seq expanded to more cell types. Second, protein-level functional validation and drug sensitivity tests should be conducted in parallel to the TIC-specific genes derived in this study. Third, clinical reproducibility should be validated using patient-derived samples, and clinical data-based tests should be carried out to evaluate the practical applicability of machine learning models. The present study comprehensively describes the transcriptomic specificity of TICs, functional core pathways, and the possibility of a predictive classification model. Thus, it is the first to provide a basis for refining tumor treatment targeting strategies and a precise TIC-management approach.

## 5. Conclusions

This study quantitatively and functionally elucidated the transcriptome-based differences between TIC and non-TIC populations. It also developed a machine learning classifier that could effectively identify TICs. The results showed that TICs have a unique gene expression signature distinct from non-TICs, including ncRNA, metabolic transporters, and ribosomal proteins. TIC-specific genes are concentrated in pathways related to survival, growth, stemness maintenance, and metabolic reorganization, which support the biological identity of TICs and their main role in cancer development. In addition, the transcriptome-based logistic regression classifier developed in this study can be used to successfully classify TICs and non-TICs with high accuracy and prediction reliability. It can also be utilized for TIC diagnosis and treatment target selection in the future. This study expanded our understanding of the molecular definition and targetability of TICs. It also provided an important basis for understanding cancer origin and treatment resistance.

## Figures and Tables

**Figure 1 cells-14-01255-f001:**
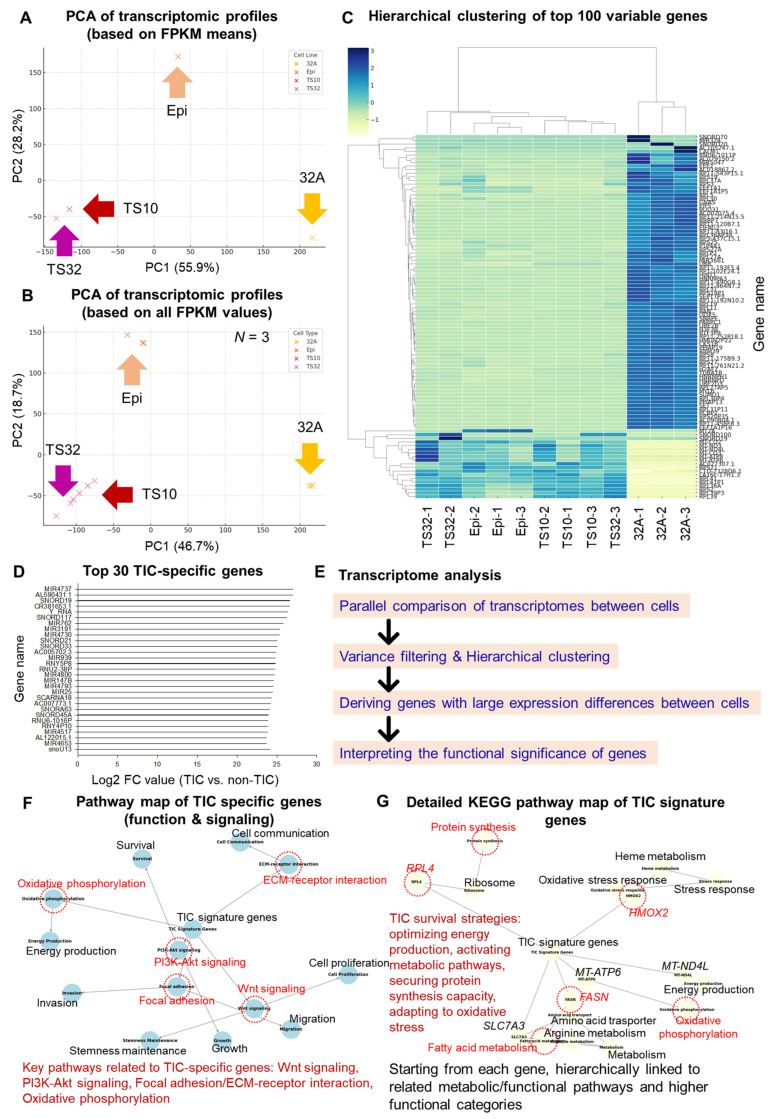
Analysis of genetic and functional differences between cell groups based on transcriptome analysis. (**A**) Principal component analysis (PCA) plot showing the transcriptomic profiles based on the values of the mean fragments per kilobase of transcript per million mapped reads (FPKM). (**B**) PCA plot showing transcriptomic profiles based on all individual FPKM values (*n* = 3). (**C**) Hierarchical clustering heatmap of the top 100 most variable genes across cell lines (all three replicates per cell line). (**D**) Bar plot showing the top 30 tumor-initiating cell (TIC)-specific genes with the highest log2 fold-change (FC) compared with non-TIC. (**E**) Overview of the analysis workflow for extracting significant gene differences and functional interpretation. (**F**) Pathway map of TIC-specific genes illustrating their functional and signaling associations. (**G**) Detailed Kyoto Encyclopedia of Genes and Genomes (KEGG) pathway map highlighting the TIC signature genes and their involvement in survival-related biological processes. The red dotted lines in figure (**F**,**G**) indicate significant nodes. Information for each node has been re-presented as text for easy visual confirmation (**F**,**G**).

**Figure 2 cells-14-01255-f002:**
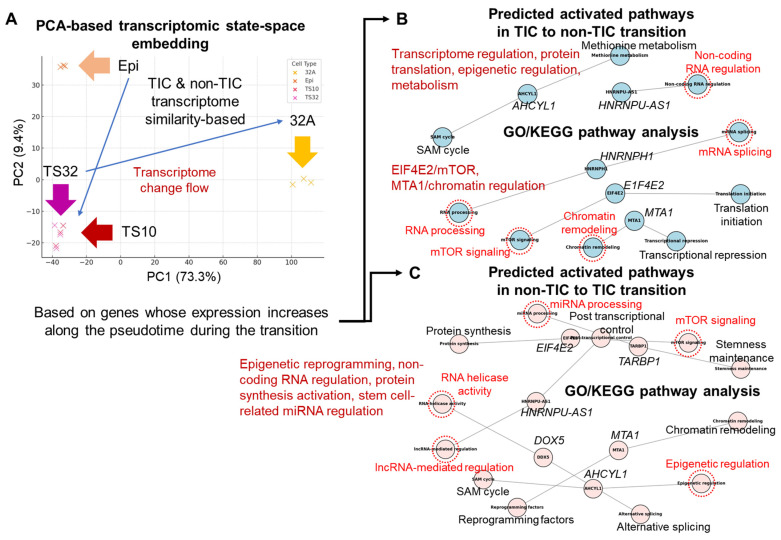
Confirming the correlation between cell groups through transcriptomic state-space embedding analysis. (**A**) Principal component analysis (PCA)-based transcriptomic state-space embedding plot illustrating the inferred differentiation flow from non-tumor-initiating cells (non-TICs) to TIC populations. (**B**) Functional enrichment analysis (Gene Ontology [GO] and Kyoto Encyclopedia of Genes and Genomes [KEGG] pathways) of genes driving the TIC-to-non-TIC transition, highlighting key biological processes and signaling pathways. (**C**) Functional enrichment analysis (GO and KEGG pathways) of genes driving the non-TIC-to-TIC transition, emphasizing key biological processes and signaling pathways. The red dotted lines in the figure (**B**,**C**) indicate significant nodes. Information for each node has been re-presented as text for easy visual confirmation (**B**,**C**).

**Figure 3 cells-14-01255-f003:**
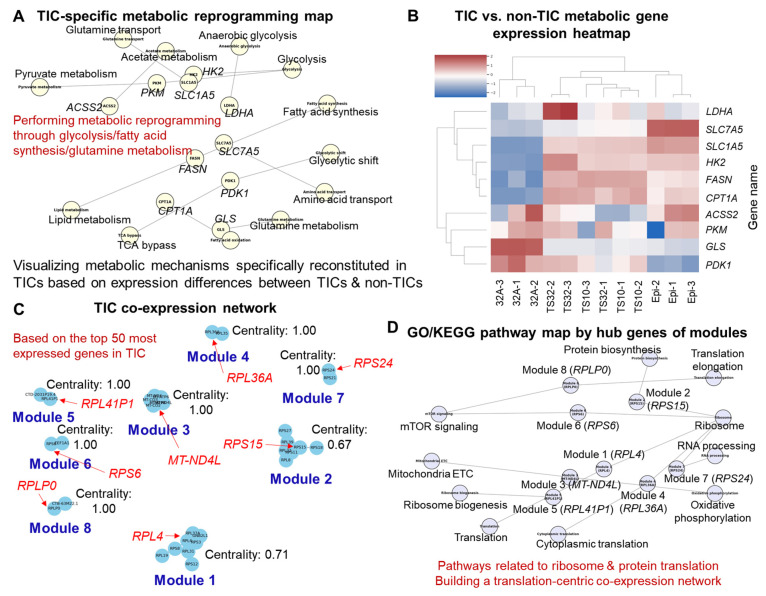
Exploring the survival mechanisms and co-expression network composition of tumor-initiating cells (TICs) based on metabolomic reconstruction. (**A**) TIC-specific metabolic reprogramming map visualizing key metabolic pathways, including glycolysis, fatty acid synthesis, and glutamine metabolism. (**B**) Heatmap comparing metabolic gene expression profiles between TIC and non-TIC groups, highlighting differentially expressed metabolic genes. (**C**) Co-expression network constructed from the top 50 most highly expressed genes in TICs, identifying major modules and their hub genes with centrality scores. (**D**) Gene Ontology (GO) and Kyoto Encyclopedia of Genes and Genomes (KEGG) pathway map highlighting the functional roles of hub genes from the co-expression modules, particularly emphasizing ribosome and protein translation-related pathways. Information for each node has been re-presented as text for easy visual confirmation (**A**,**D**). Each core module gene, based on centrality analysis, has been re-presented in text for easy visual identification (**C**).

**Figure 4 cells-14-01255-f004:**
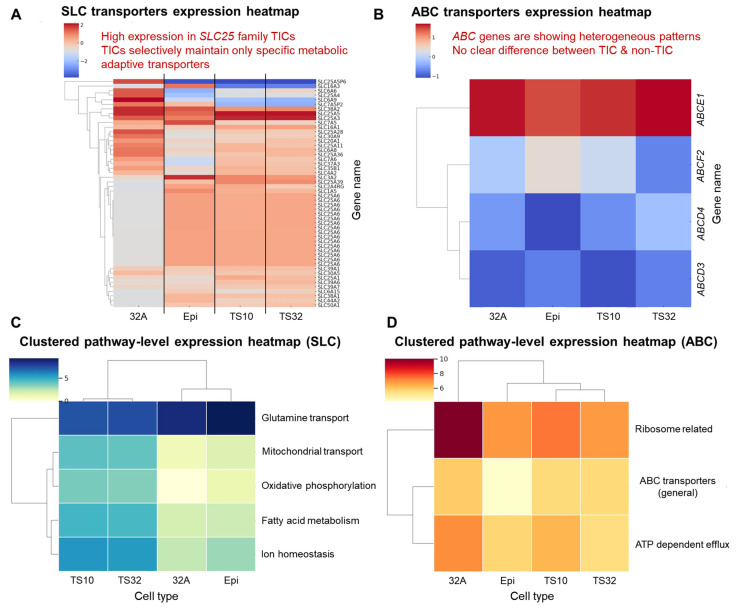
Survival and metabolic drug resistance analysis of tumor-initiating cells (TICs) with a focus on transporters. (**A**) Heatmap showing the expression of solute carrier (SLC) family transporters across TIC and non-TIC groups. Note: Multiple rows labeled as *SLC25A6* correspond to distinct entries in the reference annotation (GENCODE), each with a different Gene ID. These entries likely represent transcript isoforms or annotation redundancies and were treated as separate features in the analysis because they passed the variability filtering criteria. As a result, they appear as visually similar rows in the heatmap. (**B**) Heatmap comparing the ATP binding cassette (ABC) transporter gene expression between TIC and non-TIC groups. (**C**) Clustered pathway-level heatmap summarizing SLC-related functions such as glutamine transport, mitochondrial transport, oxidative phosphorylation, fatty acid metabolism, and ion homeostasis across cell types. (**D**) Clustered pathway-level heatmap summarizing ABC transporter-related functions, including ribosome-related processes, ABC transporting, and ATP-dependent efflux across cell types.

**Figure 5 cells-14-01255-f005:**
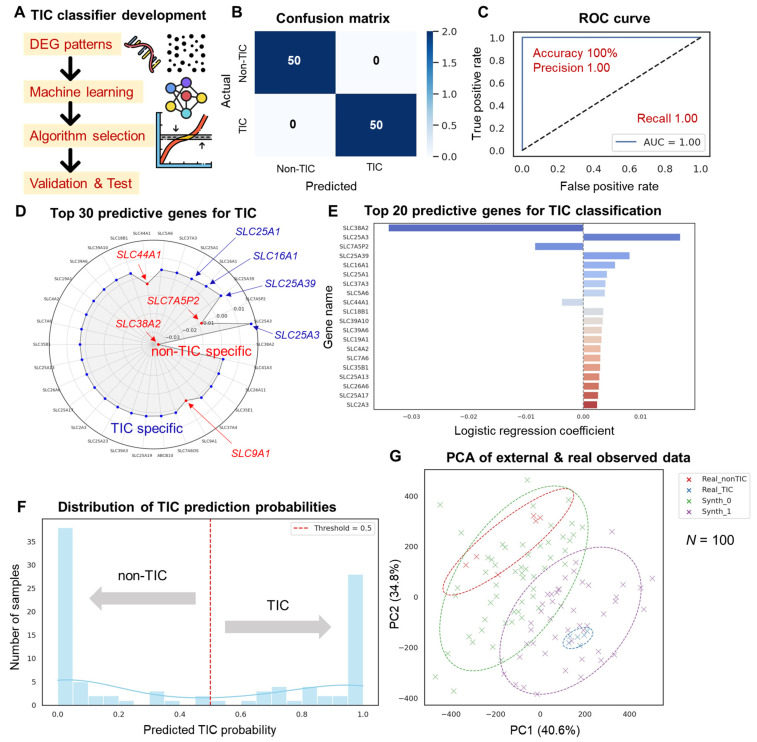
Development of a tumor-initiating cell (TIC) classifier system using machine learning. (**A**) Overview schematic illustrating the machine learning workflow, including differentially expressed gene (DEG) selection, algorithm training, and validation. (**B**) Confusion matrix showing the performance of the TIC classification model with perfect prediction. (**C**) Receiver operating characteristic (ROC) curve of the TIC classifier indicating area under the curve (AUC) = 1.00, demonstrating an excellent discriminative ability. (**D**) Radial ranking plot of the top 30 predictive genes. The radius encodes normalized importance (|β| scaled to [0, 1]) from the logistic-regression classifier; label color indicates the sign of β (blue: TIC-increasing, red: non-TIC-increasing). (**E**) Top 20 coefficients (β) shown as bars to display direction and magnitude (Panel D provides a compact ranking overview, whereas Panel E provides quantitative signed effects). (**F**) Histogram and density plot of the predicted TIC probabilities across all samples, illustrating model confidence and threshold. (**G**) Principal component analysis (PCA) plot combining the external synthetic (*n* = 100) and real data, showing clustering patterns and classification consistency with confidence ellipse. (**G**) Principal component analysis (PCA) of original (real) and external synthetic samples. “Real” denotes the original RNA-seq samples measured in this study (TIC: TS10/TS32; non-TIC: 32A/Epi; biological replicates included). “External synthetic” denotes simulated samples generated to mimic the distribution of the real data (Section 2.5).

**Table 1 cells-14-01255-t001:** Drug repositioning candidates targeting the gene expression signatures specific to tumor-initiating cells (TIC).

Gene Name	Feature Summary	TIC-Specific Expression (Corresponding Data)	Drug Examples ^c^	Drug Action Mechanism/Steps ^c^	Comprehensive Evaluation ^d^	Reference
SLC25A1	Citrate transport ^a^/Mitochondrial metabolism regulation ^b^	Figure 3B and Figure 5D–E	CTPI-2	SLC25A1 inhibition/Preclinical	Very appropriate	[35]
SLC16A1	Lactate transport ^a^/Glycolysis and pH regulation ^b^	Figure 3B and Figure 5D–E	AZD3965	MCT1 inhibition/In clinical trials	Very appropriate	[36]
FASN	Fatty acid synthesis ^a^/Survival and cell membrane composition ^b^	Figure 1G and Figure 3A	TVB-2640	FASN inhibition/Phase 2 clinical trial	Very appropriate	[18]
SLC25A3	Mitochondrial phosphate transport ^a^/ATP synthesis ^b^	Figure 5D–E	N/A	Early development	Possible	[43]
SLC25A39	Mitochondrial glutathione transport ^a^/Response to oxidative stress ^b^	Figure 5D	N/A	Oxidative stress defense/Early development	Possible	[44]
SLC7A5	Leucine transport ^a^/mTOR activation linkage ^b^	Figure 3B	JPH203	Inhibition of neutral amino acid transport/Phase 1 clinical trial	Appropriate	[16]
SLC1A5	Glutamine transport ^a^/Metabolism axis link ^b^	Figure 3B	V-9302	Glutamine analogs/Preclinical	Appropriate	[17]
LDHA	Lactate dehydrogenase ^a^/Maintaining glycolysis ^b^	Figure 3A	FX11	LDHA inhibition/Preclinical	Appropriate	[19]
HK2	Hexokinase-2 ^a^/Glycolysis rate control ^b^	Figure 3A	3-BrPA	HK2 inhibition/Preclinical	Appropriate	[20]
PKM	Glycolysis terminal enzyme ^a^/Energy production ^b^	Figure 3A	PKM2 inhibitor	Glycolysis inhibition/Exploring	Possible	[21]
GLS	Glutaminase ^a^/Metabolism reconstitution ^b^	Figure 3A	CB-839	GLS inhibition/In clinical trials	Very appropriate	[22]
ACSS2	Acetic acid → Acetyl-CoA ^a^/Fatty acid synthesis ^b^	Figure 3A	ACSS2 inhibitor	Acetyl-CoA synthesis inhibition/Exploring	Possible	[23]
RPL4	Ribosomal proteins ^a^/Translation regulation ^b^	Figure 3C–D	CX-5461	Pol I inhibition/Phase 1 clinical trial	Indirect target	[29]
HMOX2	Hemooxygenase-2 ^a^/Oxidative stress regulation ^b^	Figure 1G	Tin protoporphyrin	HMOX inhibition/Experimental	Possible	[30]
METAP1	Methionine aminopeptidase ^a^/Translation initiation ^b^	Figure 1G	N/A	Exploring	Possible	[31]

N/A: Not applicable; ^a^ Directly related to function; ^b^ Overall function related; ^c^ It was proposed by performing cross-analysis with existing drug target databases (DGIdb, DrugBank, etc.); ^d^ As evaluation indices that comprehensively reflect the validity and usability as a drug target, transcriptomic specificity in TICs, biological relevance to TIC survival/metabolism, drug targetability, and development status were comprehensively considered and evaluated.

## Data Availability

The RNA-seq datasets generated in this study are part of ongoing companion projects and are subject to an institutional data-use agreement. For this reason, the raw sequencing data and raw read-count matrix cannot be deposited in a public repository at this time. Analysis details, processed data used for figure generation (e.g., DEG lists and pathway inputs), and approach methods are available in this manuscript.

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
