# Peer review of "Interpretation of the Transcriptome-Based Signature of Tumor-Initiating Cells, the Core of Cancer Development, and the Construction of a Machine Learning-Based Classifier"

_cells, 2025, doi:10.3390/cells14161255_

Round 1
Reviewer 1 Report
Comments and Suggestions for Authors
The manuscript entitled ‘Interpretation of the transcriptome-based signature of tumor-initiating cells, the core of cancer development, and the construction of a machine learning-based classifier’ written by Ji-Hun Jang et al. presents interesting findings on transcriptomic characterization of tumor-initiating cells (TICs) and identification of markers differentiating them from non-tumor-initiating cells (non-TICs).
Through a series of bioinformatic analyses of obtained RNA-seq data, the authors identified the most significantly differentially expressed genes in two TIC cell lines compared to two non-TICs cell lines. The functional chracterization of selected genes provided important information about cellular processes potentially contributing to TICs phenotype. Machine learning modelling provides genes with the highest predictive significance in TICs classification. Finally, potential anti-cancer therapeutic agents were identified.
The research topic is clinically significant, as targeting TICs is promising, but still not sufficiently explored field in development of anti-cancer therapies.
The manuscript is scientifically sound and well-structured. The scientific rationale is clearly articulated. The manuscript has appropriate references and the English language is clear. However, the methodology part of the manuscript lacks many imortant details, and the provided figures need to be improved for better clarity and interpretability. Below I provide identified issues that require attention.
General concept comments
- The methodology in 2.1. section is described very brief, with many methodological gaps. Specifically, the source of used cell lines (commercial or developed in the authors’ laboratory, human or animal), conditions of cell culturing, name of the kit used for libraries preparation, number of technical replicates, bioinformatic software used for analysis of raw sequencing data, genome version used for sequences annotation, number of genes in final raw reads counts matrix, should be provided. Furthermore, the tools used for PCA, hierarchical clustering, differential expression analysis, functional enrichment analysis, hub genes identification, and logistic regression should be mentioned. Information on types of distances and agglomeration measures for hierarchical clustering, as well as log2 fold change and false discovery rate cut-off thresholds should also be added. All these information are required to increase the transparency and reproducibility of the study.
- In lines 252-255 the authors mentioned “In the transporter-centered analysis, the expression of SLC family genes was significantly higher in TICs than in non-TICs. Particularly, the SLC25 family was prominently activated (Figure 4A). Conversely, ABC family genes were relatively highly expressed in non-TICs (Figure 4B)”. However, in my opnion, these conclusions are too general and described with not sufficient precision. Only part of SLC25 family members exhibits higher expression in both TICs cell lines. SLC25A6 expression is lower in TS32 and TS10 when compared to Epi, as presented in Figure 4A. Furthermore, expression of ABC family genes appears to not significantly differ between TIC and non-TIC cell lines. I recommend that the authors revise all their conclusions to ensure they are rigorously supported by the data and findings presented in the study.
Specific comments
- It is not clear what criterium was used to consider RNA samples to be of a high quality. The authors properly used a NanoDrop spectrophotometer and Bioanalyzer to assess RNA quantity, purity and integrity. However, specifying the purity criteria (i.e., absorbance ratios) and RIN integrity index threshold used for sample acceptance would enhance the scientific rigor of the study.
- I have several suggestions to improve clarity and interpretability of the heatmap in the Figure 1C. Firstly, gene names on the heatmap have too small font size, which affects readability. Secondly, as the data presented in this heatmap are non-centered, used diverging palette of colors is not suitable. I recommend to use sequential palette of colors, which more properly illustrates ordered data that progress from low to high. Light colors can be used for low data values, and progressively more dark colors for high data values. Lastly, it seems that the heatmap presents means of FPKM values for each cell line. I suggest to show in this heatmap all three FPKM values obtained for individual replicates (similarly to the Figure 3B). This could significantly enhance comprehensiveness and transparency of presented data.
- The x axis annotation label should be added to Figure 1D to facilitate interpretation, and also the genes should be shown in descending order, to show the most differentially expressed genes at the top.
- The text font in all graphs presented in the study (Figures 1F, 1G, 2B, 2C, 3A, 3C, 3D) should be enlarged to ensure the plots are readable.
- Line 198 “…the top 100 highly variable genes...” – please clarify what parameter/criterion was used to select the most variable genes.
- Line 331 “…the top 50 most actively expressed genes in the 331 TIC population…” - please clarify what parameter/criterion was used to select the most actively expressed genes.
- Please revise red font comments present in many figures. They are sometimes inconsistent (Figure 4B). Please ensure if they need to be located in the figures or maybe a better way is to place them in the corresponding figure legends.
- Figure 5B – the numbers of samples assigned to each squares in confusion matrix schould be added to make the plot more complete.
- The human gene symbols should be written in italic font, in accordance with formatting standards.
- In the Data Availability Statement the authors mentioned that “All data and related materials are accessible in this manuscript”; however, RNA-seq data were not included in the manuscript files. Therefore, I encourage the authors to share used dataset (raw reads counts matrix) as, e.g., a supplementary material or to deposit it in publicily available repository. Making the data publicly available supports open scientific exchange and meets the TOP (Transparency and Openness Promotion) guidelines, which are a standard in publishing and funding. If the data cannot be made publicly available, please provide a specific justification. In this situation, making the data accessible upon reasonable request to qualified researchers would appear an optimal solution.
I believe that my suggestions will help the authors to improve the quality of the manuscript.
Author Response
Please see the attachment.
Pages 3 through 15 of the attached "Point-by-point response" file contain comments and responses to Reviewer 1.

Reviewer 2 Report
Comments and Suggestions for Authors
The manuscript by Jang and colleagues carried out bioinformatics analysis to characterize the expression profile of tumor-initiating cells (TICs) and built a machine learning based model to predict the TIC status. Specifically, the authors collected RNA-seq transcriptomes from multiple human cell lines of TIC and non-TIC, and performed differential expression analysis and functional enrichment analysis to identify the key genes and pathways that distinguish TICs. Based on the differential genes, they constructed a TIC classifier using logistic regression and were able to validate the performance on external data. They uncovered several metabolic and translational reprogramming functions and epigenetic regulators that are unique to TICs.
Comments and suggestions:
- Line 81-82, “we performed multi-layered principal component analysis (PCA)…” It is not very clear what “multi-layered” PCA is. Also, in the following Line 90, “multi-layered analysis … were combined”. The term “multi-layered” sounds unspecific and the authors are recommended to better clarify it.
- Section 2.1 “Sample preparation and RNA-seq data acquisition”. The description regarding how raw reads were preprocessed, how reads were aligned to the reference genome, what genome build and the transcriptome annotation were used, and how gene counts were quantified is lacking. The authors need to include the data processing methods.
- Line 133-137, the pseudotime trajectory needs more theoretical foundation. Basically, trajectory analysis is often performed on time course data such as single-cell developmental transcriptomes. Here, there is no assumption that one cell line transforming to another is a “trajectory” over the time. Moreover, only 3 replicates per cell line will not make smooth transitions to form a “trajectory”. So, I would recommend the authors not to use the term “trajectory analysis” as what Figure 2A depicts is neither pseudotime nor trajectory.
- Section 2.3 “Transcriptome-wide differential expression analysis” (Line 139-143). What differential expression method did the authors use? What foldchange and significance thresholds were used? “…the association of TIC-specific genes with survival-, differentiation-, and metabolism-related pathways was evaluated.” Please detail the list of genes in the “survival-, differentiation-, and metabolism-related pathways”.
- Section 2.4 “Co-expression network and hub gene identification”. How was the network constructed? Was it based on gene-gene correlation metrics or Bayesian networks? These critical details are needed in the Methods section.
- Figure 1F-G, how significant are the pathways enriched in TIC specific genes? It would be better if the significance could be shown as the color or thickness of the edges.
- Section 3.2 “Functional enrichment and pathway analysis” (Line 271-303). Again, it is important that the statistical significance is shown in either the figures or supplementary tables. In the current manuscript, it is difficult to see how much fold of enrichment the metabolic pathways are engaged in the TIC vs non-TIC difference.
- Line 416-425, “…were randomly generated on the basis of the statistical distribution of the real data”. What statistical distribution was it based on? How many total counts does each simulated sample have? What is the sample size and how variable are the replicates? The authors are expected to fill in the missing information.
- Figure 5D-E. It is unclear what the radius means in panel D. Also, what is the difference between D and E except for top 30 genes in D and top 20 genes in E?
- Figure 5G, the legend show “PCA of external & real observed data”. But what is the “real observed data”? How is it related to the “external data” (“external synthetic”, Line 385)? Despite the good performance (AUC = 1.0, Figure 5), this part appears very confusing. More clarification is expected here.
Author Response
Please see the attachment.
Pages 16 through 29 of the attached "Point-by-point response" file contain comments and responses to Reviewer 2.

Round 2
Reviewer 1 Report
Comments and Suggestions for Authors
I appreciate the efforts of the authors to respond to my comments. I have only one minor observation: the heatmap presented in Figure 4A contains multiple rows labeled as SLC25A6. This is potentially confusing, as a single mean expression value per group for individual gene is typically expected. Please clarify (perhaps in the figure legend) the reason for the presence of multiple, very similar expression values for SLC25A6 across the studied groups. I have no further remarks.
Author Response
Please see the attachment. A response to reviewer 1's comment is provided on pages 3-4 of the attached file.
Comments from the reviewer 1:
I appreciate the efforts of the authors to respond to my comments. I have only one minor observation: the heatmap presented in Figure 4A contains multiple rows labeled as SLC25A6. This is potentially confusing, as a single mean expression value per group for individual gene is typically expected. Please clarify (perhaps in the figure legend) the reason for the presence of multiple, very similar expression values for SLC25A6 across the studied groups. I have no further remarks.
Response to the reviewer 1:
We deeply appreciate reviewer 1’s interest in our study. As you can see from the response to the reviewer comments, we have faithfully responded to the reviewer comments and carefully reviewed and improved the manuscript. We sincerely hope that reviewer 1 understands the authors’ efforts. We have revised this manuscript to reflect the reviewer 1's comments. As reviewer 1 already acknowledged, this study is an important one, significantly supplementing the molecular biological and functional characterization of TICs, a field that remains largely understudied, and providing interesting insights. We hope reviewer 1 fully appreciates the scientific significance of this study and the authors' dedicated efforts.
We thank the reviewer for pointing out the repeated SLC25A6 labels in Figure 4A. Upon re-checking the raw FPKM expression matrix, SLC25A6 appears as two distinct entries, each with a unique Gene ID (e.g. ENSG00000169100.8 and ENSGR0000169100.8) in the GENCODE reference annotation. These likely correspond to transcript isoforms or redundant annotation records. In our analysis pipeline, features were defined at the Gene ID level, and both entries passed the filtering criteria (FPKM threshold and variability selection), so they were retained as separate features in the heatmap. As their expression profiles across samples are highly similar, they appear as nearly identical rows. We have clarified this in the revised figure legend to avoid confusion.
As a result, by reflecting the reviewer 1’s comments, we revised the previous captions as follows: Survival and metabolic drug resistance analysis of tumor-initiating cells (TIC) with a focus on transporters. (A) Heatmap showing the expression of solute carrier (SLC) family transporters across TIC and non-TIC groups. Note: Multiple rows labeled as SLC25A6 correspond to distinct entries in the reference annotation (GENCODE), each with a different Gene ID. These entries likely represent transcript isoforms or annotation redundancies, and were treated as separate features in the analysis because they passed the variability filtering criteria. As a result, they appear as visually similar rows in the heatmap. (B) Heatmap comparing the ATP binding cassette (ABC) transporter gene expression between TIC and non-TIC groups. (C) Clustered pathway-level heatmap summarizing SLC-related functions such as glutamine transport, mitochondrial transport, oxidative phosphorylation, fatty acid metabolism, and ion homeostasis across cell types. (D) Clustered pathway-level heatmap summarizing ABC transporter-related functions, including ribosome-related processes, ABC transporting, and ATP-dependent efflux across cell types. (lines of 389-400, page of 12, in revised R2 manuscript)

Reviewer 2 Report
Comments and Suggestions for Authors
The authors have revised the manuscript and addressed my concerns. I do not have more questions now.
Author Response
Please see the attachment. Responses to reviewer 2's comments are provided on page 5 of the attached file.
Comments from the reviewer 2:
The authors have revised the manuscript and addressed my concerns. I do not have more questions now.
Response to the reviewer 2:
We deeply appreciate reviewer 2’s interest in our study. By fully incorporating reviewer 2's previous comments into the manuscript, we were able to significantly improve the quality of the manuscript.
